# Steady expression of high oleic acid in peanut bred by marker-assisted backcrossing for fatty acid desaturase mutant alleles and its effect on seed germination along with other seedling traits

**Sandip K. Bera**[1]*, **Jignesh H. Kamdar**[1], **Swati V. Kasundra**[1], **Sahil V. Patel**[1], **Mital D. Jasani**[1], **A. K. Maurya**[1], **P. Dash**[1], **Ajay B. Chandrashekar**[1], **Kirti Rani**[1], **N. Manivannan**[2], **Pasupuleti Janila**[3], **Manish K. Pandey**[3], **R. P. Vasanthi**[4], **K. L. Dobariya**[5], **T. Radhakrishnan**[1], **Rajeev K. Varshney**[3]

1 Indian Council of Agricultural Research-Directorate of Groundnut Research (ICAR-DGR), Junagadh, India, 2 National Pulses Research Center, Tamil Nadu Agricultural University (TNAU), Vamban Colony, Pudukkottai, Tamil Nadu, India, 3 International Crops Research Institute for the Semi-Arid Tropics (ICRISAT), Hyderabad, India, 4 Regional Agricultural Research Station, Acharya NG Ranga Agricultural University (ANGRAU), Tirupati, India, 5 Main Oilseeds Research Station, Junagadh Agricultural University (JAU), Junagadh, India

* berask67@yahoo.co.in

## Abstract

Peanut (*Arachis hypogaea* L.) is an important nutrient-rich food legume and valued for its good quality cooking oil. The fatty acid content is the major determinant of the quality of the edible oil. The oils containing higher monounsaturated fatty acid are preferred for improved shelf life and potential health benefits. Therefore, a high oleic/linoleic fatty acid ratio is the target trait in an advanced breeding program. The two mutant alleles, *ahFAD2A* (on linkage group a09) and *ahFAD2B* (on linkage group b09) control fatty acid composition for higher oleic/linoleic ratio in peanut. In the present study, marker-assisted backcrossing was employed for the introgression of two *FAD2* mutant alleles from SunOleic95R into the chromosome of ICGV06100, a high oil content peanut breeding line. In the marker-assisted backcrossing-introgression lines, a 97% increase in oleic acid, and a 92% reduction in linoleic acid content was observed in comparison to the recurrent parent. Besides, the oleic/linoleic ratio was increased to 25 with respect to the recurrent parent, which was only 1.2. The most significant outcome was the stable expression of oil-content, oleic acid, linoleic acid, and palmitic acid in the marker-assisted backcrossing-introgression lines over the locations. No significant difference was observed between high oleic and normal oleic in peanuts for seedling traits except germination percentage. In addition, marker-assisted backcrossing-introgression lines exhibited higher yield and resistance to foliar fungal diseases, *i.e.*, late leaf spot and rust.

**Data Availability Statement:** All relevant data are within the manuscript and its Supporting Information files.

**Funding:** The work presented in this article is a contribution from research projects sponsored by Integrated Scheme of Oilseeds, Pulses, Oilpalm and Maize (ISOPOM), Department of Agriculture and Co-operation (DAC), Ministry of Agriculture, Government of India.

**Competing interests:** All authors have contributed equally to this research work and have no competing interests exist.

## Introduction

Peanut or groundnut (*Arachis hypogaea* L.) is one of the world's most important legumes for its valuable edible oil and protein content. It is a major cash crop and plays an essential role in the livelihood of millions, especially in semi-arid tropics. It is cultivated globally in around 27.94 million ha with a total production of 47.09 million tons [1]. China, India, Nigeria, and the United States of America are the leading groundnut producers that account for ~70% of the global peanut production. Peanut is traditionally used for the extraction of oil for edible as well as industrial purposes but the quality attributes vary with geographical region. In China, India, and other Asian countries, half of the produce is crushed for oil extraction and the rest is being used for confectionary and food purposes. While in the USA and other European countries more than two-thirds of peanut production are used for confectionary and food purposes and remaining one-third is used in the extraction of oil. Low oil content peanuts are preferred for table purposes and other food preparations of low caloric value.

Different proportions of saturated fatty acids (SFAs), monounsaturated fatty acids (MUFA) and polyunsaturated fatty acids (PUFA) determine the nutritional quality, shelf life, and flavor of peanut oil as well as its products. The peanut oil contains 80% unsaturated fatty acids (UFAs), mainly oleic (MUFA), and linoleic (PUFA) acids, whereas the remaining 20% SFAs comprises of palmitic, stearic, arachidic, behenic and lignoceric acid. Palmitic acid alone contributes half of the total SFAs while the rest five make up the remaining 50% [2]. SFAs are considered to increase serum low-density lipoproteins cholesterol level in the blood [3]. An elevated level of palmitic acid in the oil also increases the risk of cardiovascular diseases (CVD) [4]. A higher proportion of linoleic acid results in off flavors, rancidity, the short shelf life of oil and its derived products, which makes it undesirable for cooking purpose [5]. From a nutritional point of view, MUFA have been desirable in lowering plasma cholesterol levels and reduced risk of CVD [6, 7]. Therefore, a diet with high oleic (HO) acid can reduce the risk of heart diseases, inflammatory diseases tumorigenesis, and slow down atherosclerosis [8, 9]. In addition, oleic acid has ten-fold higher auto-oxidative stability than linoleic acid [10]. Therefore, there is a greater demand for the improved lines with higher oleic/linoleic (O/L) ratio in the peanut oil.

In peanut, fatty acid desaturase enzyme catalyzes desaturation of oleic to linoleic acid. [11, 12]. It is controlled by two homeologous genes *ahFAD2A* and *ahFAD2B*, located on A-genome (linkage group a09) and in B-genome (linkage group b09), respectively [13,14]. Mutations in *ahFAD2A* and *ahFAD2B* genes results in reduced fatty acid desaturase enzyme activity that leads to higher accumulation of oleic acid [13,15]. A single base pair (bp) substitution mutation (G:C to A:T) in *ahFAD2A* gene at 448 bp position results in a missense amino acid from aspartic acid to asparagine (D150N). While, an insertion mutation in A:T of *ahFAD2B* gene at 442 bp position generates premature stop codon [11, 12]. Thus the two mutant fatty acid desaturase alleles stop the conversion of oleic acid to linoleic acid in peanut [16, 17, 18]. Improved breeding lines with HO and lower linoleic and palmitic acids in peanut oil are essential to make peanut of superior quality. Norden *et al.,* [19] first identified F435 as a natural peanut mutant line with approximately 80% oleic acid and 2% linoleic acid. Later on, the first ever HO peanut breeding line, SunOleic95R, was produced with the help of conventional breeding method in the USA [16]. Chen *et al.,* [20] and Chu *et al.,* [13] developed linked allele specific-polymerase chain reaction (AS-PCR) and cleaved amplified polymorphic sequence (CAPS) markers, respectively for both of the *ahFAD2* alleles. The development of the associated markers in peanut helped in the improvement of 'Tifguard High O/L' variety in the USA through marker-assisted backcrossing (MABC) [21]. Recently, Janila *et al.,* [22] introgressed *ahFAD2* alleles from SunOleic95R into the elite breeding lines using MABC and marker-assisted

selection (MAS) at ICRISAT, Patancheru, India. Further, Bera *et al.*, [23] developed HO peanut lines through MAS at ICAR-Directorate of Groundnut Research, Junagadh, India. Most of these molecular breeding lines are under examination in All India Coordinated Research Project on Groundnut (AICRP-G) and, recently, Girnar 4 and Girnar 5 genotypes have been identified for release in India.

Peanut is grown in both rainy and post-rainy (as winter and summer crop) seasons across different states of India, varying largely in climatic and edaphic conditions. The chemical composition of peanut oil is influenced by several factors like genotype, geographic location, season, soil humidity, temperature and growing conditions [24, 25,26]. In general, lower temperature (22-29˚C) is associated with more linoleic acid synthesis due to increased activity of oleate desaturase enzyme [27, 28] and high temperature (30-33˚C) during pod filling to harvesting stage reduces the linoleic acid content in peanut oil [29, 30, 31]. Li *et al.*, [32] also reported that season and temperature had a significant influence on fatty acid content in Brassica crops. Flagella *et al.*, [33] reported a reduction in oleic and stearic acid while an increase in linoleic and palmitic acid in sunflower under irrigated cultivation. Furthermore, healthy and vigorous seedlings are one of the important criteria for making HO peanut cultivation profitable. The chemical composition of seed reserve might affect its germination and seedling vigor as seed reserve content is correlated with germination percentage [34]. In oilseeds, the major storage reservoir is lipid that provides essential energy to the growing embryo and thus affects seed germination. The alterations in seed lipid affect membrane lipid composition in respect to membrane function and permeability, which affects germination, vigor, and tolerance to environmental stress [35]. In peanut, germination percentage decreases with increase in O/L and unsaturated/saturated fatty acid ratios especially at lower (16˚C and 14˚C) temperatures [36]. Sun *et al.*, [35] found that seed vigor of high oleate lines was lower as compared with the lines with normal oleic content in peanut. Upadhyaya *et al.*, [37] reported a poor yield of ICG-2381, a groundnut accession with high O/L ratio.

Considering the demand of peanut with HO both in domestic and international markets, the present study was undertaken with three objectives: i) introgression of *ahFAD2* alleles into the higher oil content peanut variety through MABC; ii) multi-location testing of MABC derived HO peanut lines over the two seasons for yield and impact of locations and seasons on the oil quality and oil content iii) determining the effect of HO trait on seed germination and other seedling traits.

## Materials and methods

### Plant material

For improving the oil quality, ICGV06100 was used as female/recurrent and SunOleic95R as male/donor parents for MABC breeding program. ICGV06100 is a high yielding and high oil containing (~55%) peanut line but with lower oleic acid (~39.3%), developed by ICRISAT, Patancheru, India (ICRISAT, 2012; unpublished). It is a Virginia bunch (semi-spreading) cultivar derived from the cross [(ICGV92069 × ICGV93184) × (NCAc-343 × ICGV86187) × S23]. SunOleic95R, having both *ahFAD2* mutant homozygous alleles with HO (~80%) but lower yield and oil contents (~45%) was used as a male/donor parent. It was developed by Florida Experimental Agriculture Station, USA, from the mutant line F435 [16].

Under the second objective, MABC lines were tested for pod yield in multiple seasons and locations. Initial yield evaluation of MABC lines along with elite cultivars (Abhaya, CO-6, GG-20, ICGS-1043, GPBD-4, JL-24, TMV-2, VRI-6, K-6, TAG-24, GJG-31, and TG-37A) was done at a single location over two seasons. Subsequently, advanced yield evaluation of MABC lines together with other breeding lines and elite cultivars was done at three different locations.

Besides fatty acids profile, oil and protein contents of MABC lines, SunOleic95R and ICGV06100 were also estimated at three locations.

For the accomplishment of the third objective, two separate panels of peanut genotypes were studied for seed and seedling traits. The first panel consisted of normal oleic acid peanut genotypes (GG-20, ICGV06100, ICGV05141, and ICGV06110), while the second panel had HO peanut genotypes (NRCGCS-587, HOP-IL_MAS-191, HOP-IL_MAS-145and HOP-IL_MAS-130) [22, 23, 38].

## Molecular markers

Two types of DNA-markers linked to *ahFAD2* mutant alleles were used for genotyping. The allele specific-polymerase chain reaction (AS-PCR) markers [20] were used to identify heterozygous plants for the mutant alleles. The cleaved amplified polymorphic sequence (CAPS) markers [13] were deployed to select homozygous plants for both the *ahFAD2* alleles.

## DNA extraction and marker genotyping

The DNA was extracted from tender fresh leaves of 10 to 15 days old field-grown seedlings using modified cetyltrimethylammonium bromide (CTAB) extraction method [39]. The quality and quantity of DNA were checked [25] and genotyping of the target population was done using AS-PCR and CAPS markers. The primer combination, F435-F and F435SUB-R, amplified 203bp fragment for the mutant allele (substitution from G:C→A:T, *ahFAD2A*) in the A-genome, while the primer combination, F435-F and F435INS-R amplified 195bp fragment for the mutant allele (A:T insertion, *ahFAD2B*) in the B-genome (Fig 1). In case of wild type *ahFAD2A* allele, the 826bp fragment was digested to 598bp and 228bp, while the mutant genotypes had the 826bp fragment intact. For B-genome, 2.0U of restriction enzyme Hpy188I (New England Biolabs, UK) was used for digestion of 10μl of PCR amplicon for about 16 hours at 37˚C. The wild type *ahFAD2B* allele of 1214bp with five restriction sites cleaved into five fragments i.e., 736, 263, 171, 32 and 12bp.While the mutant allele had one additional restriction site in the 736bp fragment which was further cleaved into 550 and 213bp (all together six restriction sites in mutant instead of five in wild type) [23, 25].

## Estimation of background genome recovery and linkage drag

Eighty polymorphic single sequence repeats (SSRs) from 20 linkage groups (preferably two from each arm of a linkage group) were deployed to determine recurrent parent genome recovery in MABC lines [40, 41]. Furthermore, recurrent parent and MABC lines were assessed based on the passport data. Subsequently, the desirable recombinant plants possessing the smallest size of introgressed segments with minimum linkage drag among MABC lines were identified. For the analysis, additional 10 SSRs, selected from the ~20cM genomic region on either side of *ahFAD2* loci from both a09 and b09 linkage groups, were used (S1 Table).

## Hybridization and development of MABC lines

Hybridization was done at ICRISAT, Patancheru, India in 2011 during the rainy season. The crossed seeds were planted at ICAR-DGR, Junagadh in post-rainy season in the same year. $F_1$s were genotyped with linked allele-specific markers to identify true $F_1$ plants and plants heterozygous for *ahFAD2* alleles were used for backcrossing. The $BC_1F_1$ plants were planted in 2012 rainy season and were genotyped with allele-specific markers to identify heterozygous plants at both the loci. Backcrossing and genotyping with AS-PCR markers were continued until the development of $BC_3F_1$ generation. The $BC_3F_1$ seeds were planted in 2013 rainy season and

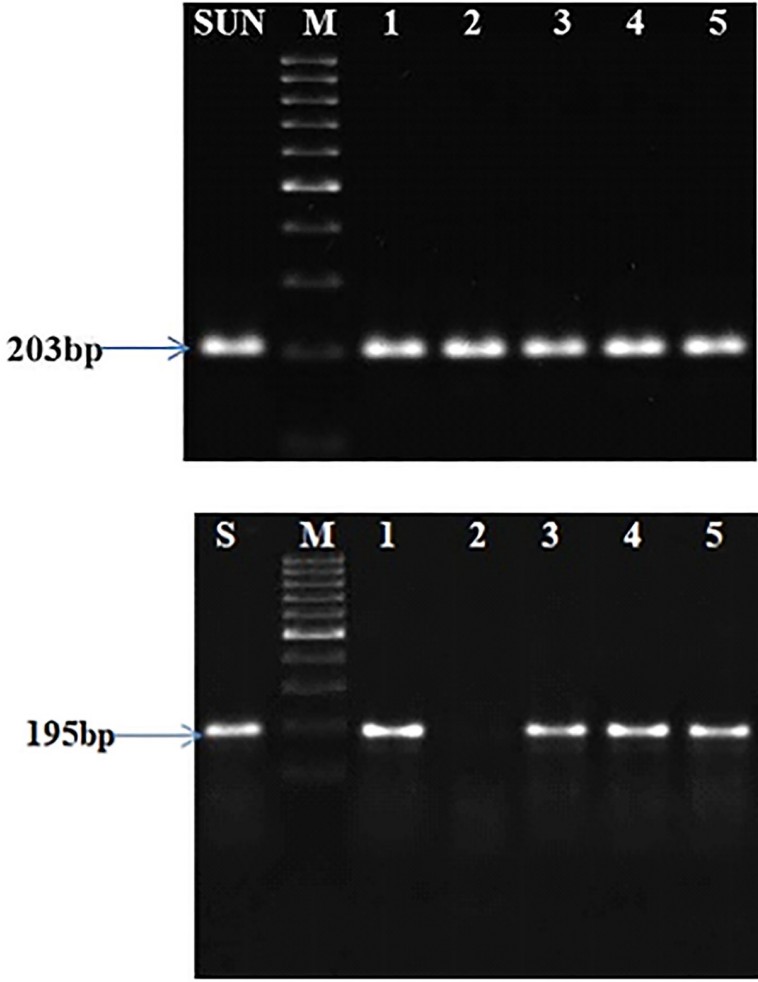

**Fig 1.** AS-PCR assay, (a) Amplification of *ahFAD2A* mutant allele-specific 203 bp amplification in 1 to 5 $F_1$ plants; (b) *ahFAD2B* mutant allele-specific 195 bp in3 to 4 while absent in 1 and 2 $F_1$ plants; where SUN: SunOleic95R, M:100bp DNA ladder.

plants having *ahFAD2* alleles were advanced to $BC_3F_2$ generation. The $BC_3F_2$ seeds were planted in 2013 post-rainy season and plants were genotyped with CAPS marker to identify plants with both the homozygous mutant loci. The $BC_3F_{2-3}$ plants homozygous for *ahFAD2* alleles were advanced to $BC_3F_{3-4}$ in 2014 rainy season. Phenotyping for oil content and fatty acid composition was done in $BC_3F_{3-4}$progeny. Finally, introgression lines (ILs) were selected based on oleic acid content and was coded as MABC introgression lines (MABC-ILs).

## Yield evaluation of MABC-ILs lines

The initial yield evaluation of MABC lines along with elite peanut cultivars was done in 2014 post rainy and 2015 rainy seasons. In both the seasons, genotypes were planted in a randomized block design (RBD) with three replications. The advanced yield evaluation of MABC-ILs along with other breeding lines and elite cultivars was carried out at three different states, namely Gujarat, Telangana, and Andhra Pradesh in both 2016 rainy and 2016 post-rainy seasons. The crops were sown in RBD with two replications. Each genotype was planted on four-meter beds in four lines. Recommended crop management practices were followed for raising

a healthy crop. Pod yield per plot (7.2 m$^2$) was recorded during the harvest on maturity of crop (111–115) days after sowing.

## Biochemical analysis for oil content and fatty acid profile

The harvested mature kernels were subjected to oil and fatty acid analysis using Gas chromatograph (model number GC-700, Thermo Fisher, USA) [42] with flame ionization detector (FID) [23].

## Seed and seedling traits

The matured kernels harvested from the plants of rainy season 2018 were subjected to the analysis. The pods harvested in the first week of October 2018 were sown in the third week of February 2019. The experiment followed RBD and was conducted in a BOD incubator (San-134, Sanco) under controlled temperature (32 ±2˚C), humidity (70 ±5%), and cooled LED lights for 24 h. Each genotype was sown in five replications with 20 kernels per replication in randomized complete block design (RCBD). Ten kernels were sown in a UV protected 7×8 inch black-color plastic plant nursery bags, filed with normal soil (~2.3kg). Thus, two plastic plant nursery bags constituted single replication. The kernels were treated with Bavistin$^®$ (2 g per kg of kernels) prior to sowing. After sowing, watering was done until saturation of the polythene bags and kept in BOD for 15 days. Regular watering was maintained on every alternate day. Plastic bags were removed carefully after 15 days so that there was no damage to the root system. The individual plant was collected replication wise from each genotype after thorough washing (Fig 2). Observations on the rate of germination, shoot length, root length, shoot fresh weight, root fresh weight, plant dry weight, root dry weight, and vigor index were recorded. The rate of germination was calculated using the formula: Germination (%) = (number of seeds germinated/total number of seeds sown) × 100. Vigor index was calculated using the formula: Vigor Index = (Seedling dry weight× germination %) /100 [43].

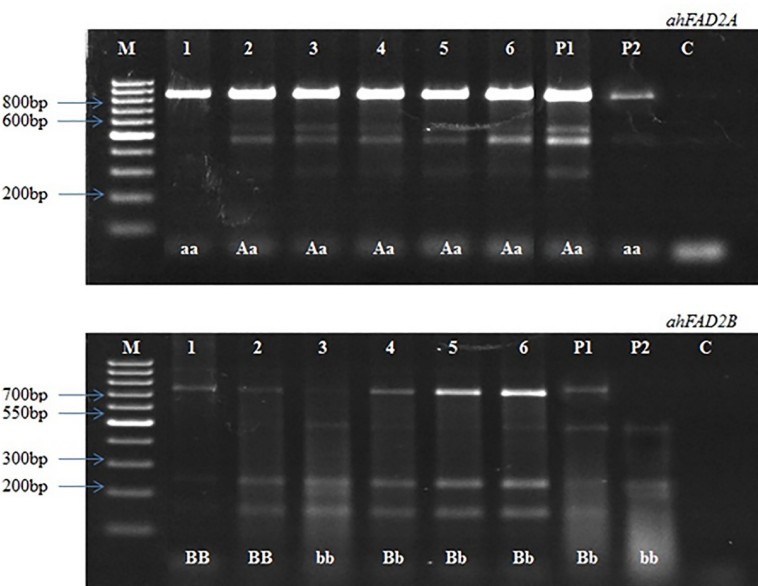

**Fig 2.** Groundnut genotypes grown in BOD; a) Plants grown in polythene bags, b) Plants uprooted for recording observations on seedling traits.

## Characterization of genotype

The passport data of MABC-ILs and recurrent parent were recorded on the basis of 16 qualitative and 17 quantitative traits, along with 6 special features, following peanut-descriptor [44] from five plant samples collected from the field at vegetative, reproductive, and harvesting stages.

## Statistical analysis

Recurrent parent genome (RPG) recovery was analyzed using the formula: "RPG% = [{2 (R) + (H)}/2N] × 100" [45]; where "R" is the number of loci homozygous for recurrent parent allele; "H" is the number of loci still remaining heterozygous, and "N" is the total number of polymorphic markers used in the background analysis. The stability analysis for the pod yield was performed using AMMI ANOVA and GGE biplot models using R package [46]. A t-test was applied to assess the mean difference between oil, protein, moisture, oleic acid, linoleic acid, and palmitic acid contents among the MABC-ILs and parents. The significant differences between the mean values were determined by Duncan's multiple range test (DMRT) (Duncan 1955) at a $P \leq 0.05$ using CropStat version 7.2 [47]. Significant differences if any, between genotypes were compared using ANOVA.

# Results

## Development of advanced ILs through MABC

The crossed seeds received from ICRISAT, Patancheru, were planted at ICAR-DGR, Junagadh and resulted in 15 $F_1$ plants. Eight plants were identified as true hybrids carrying both the mutant $aFAD2$ alleles. These eight $F_1$ plants were used as pollen parents to make the first backcross with the recurrent parent. Out of 28 $BC_1F_1$ plants, six plants were found to carry both the $ahFAD2$ alleles in a heterozygous condition. Second backcrossing resulted in 32 $BC_2F_1$ plants and both the mutant alleles were found in nine plants. Third backcrossing resulted in 37 $BC_3F_1$ plants, among which six plants carried the $ahFAD2$ alleles. These six $BC_3F_1$ plants were selfed and 67 $BC_3F_2$ seeds were harvested and sown in the next season. $BC_3F_2$ plants were genotyped with the AS-PCR and CAPS markers, and three plants were finally identified as homozygous for both the $ahFAD2$ alleles (Fig 3). Subsequently, the fatty acid analysis confirmed single MABC-IL with ~80% oleic acid (which was later coded as NRCGCS-587).

## Recurrent parent genome recovery and linkage drag

Eighty SSRs were polymorphic between the recurrent parents and NRCGCS-587. Homozygosity was found with 73 SSRs in NRCGCS-587 indicating 91.87% recurrent parent genome (RPG) recoveries. However, a genomic segment carrying the $ahFAD2$ alleles was present in NRCGCS-587. Out of the 10 polymorphic SSRs tested between SunOleic 95R and NRCGCS-587, nine SSRs were amplified only in NRCGCS-587 and not amplified in SunOleic 95R (S1 Table) indicating a linkage drag of additional segments away from the $ahFAD2A$ and $ahFAD2B$ alleles. Therefore, introgression of additional genomic regions in NRCGCS-587 resulted in some linkage drag but it showed no decrease in high oleic content.

## Fatty acid profile analysis and estimation of oil content in MABC-IL (NRCGCS-587) and parents

Fatty acid profile analysis of NRCGCS-587 with its parents was done in two seasons (S2 Table). In 2014 post-rainy season plantations, oleic acid and linoleic acid contents in

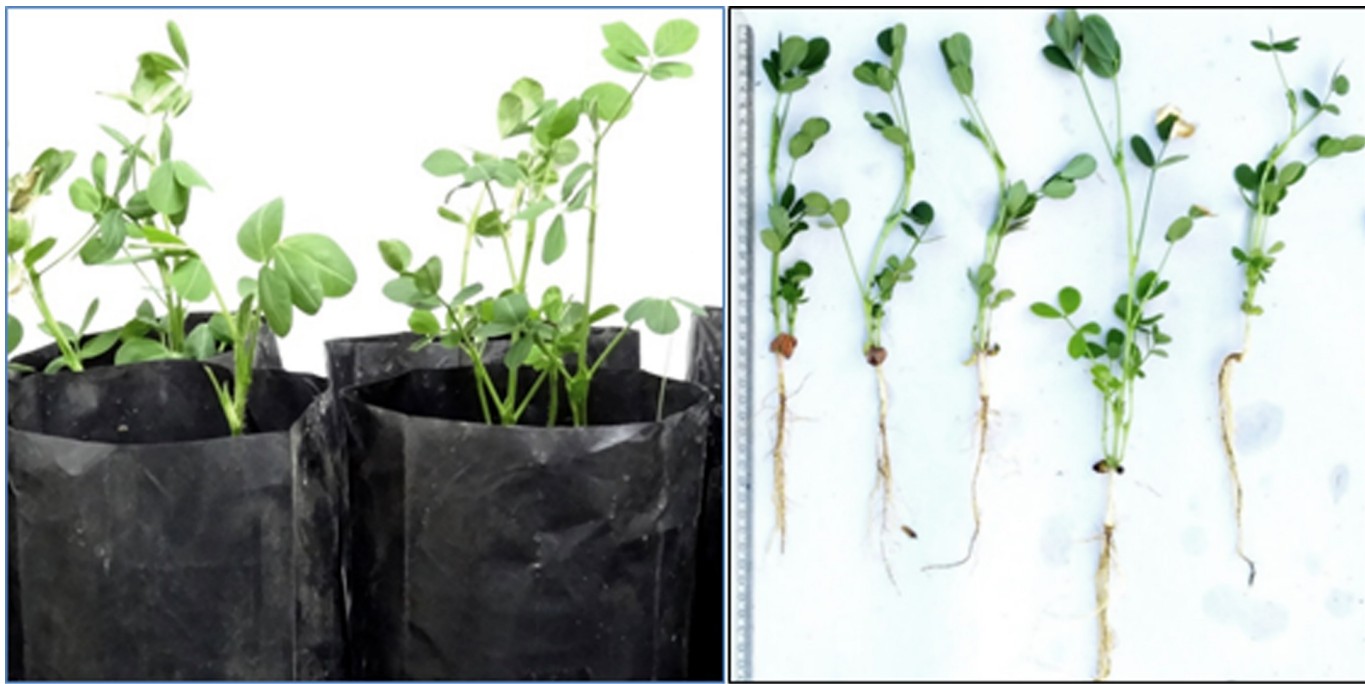

**Fig 3.** CAPS assay; (a) Heterozygous and homozygous plants for *ahFAD2A* mutant allele; (b) Heterozygous and homozygous plants for *ahFAD2B* mutant allele; where M: 100bp DNA ladder, 1–6: MABC-ILs, P1: ICGV06100, P2: SunOleic95R, C: Control, 'AA, BB': homozygous wild alleles, 'Aa, Bb': heterozygous alleles and 'aa, bb': indicates homozygous mutant alleles.

NRCGCS-587 were recorded as 78.8% and 4.0%, respectively. Whereas the same were 42.0% and 35.0% in the recurrent parent, respectively, and as 77.0% and 6.0% in the donor parent, respectively. The O/L ratio in NRCGCS-587 was 19.7, while it was 1.2 in the recurrent parent. The palmitic acid content was 6.8% in NRCGCS-587 as compared to 13.0% and 7.0% in the recurrent and donor parent, respectively (Fig 4). NRCGCS-587 contained 53% oil and 24% protein as compared to 54% oil and 26% protein in the recurrent parent and 48% oil and 26% protein in the donor parent (Fig 5). Further analysis of the oil content and fatty acid composition was done in 2015 rainy season. NRCGCS-587 showed 54% oil and 23% protein content, ICGV-06100 contained 54% oil and 24% protein, and SunOleic95R recorded 50% oil and 25% protein contents. So, there was no significant differences in oil and protein content of NRCGCS-587 with its parents. (Fig 5). In NRCGCS-587, oleic acid, linoleic acid, and palmitic acid contents were 81%, 3%, and 6%, respectively, as compared to 39%, 39%, and 9% in ICGV06100, and 80%, 3.0%, and 6.0%, in SunOleic95R, respectively (Fig 4). The O/L ratio was 27.0 in NRCGCS-587, while it was 1.0 in the recurrent parent and 23.25 in the donor parent.

## Pod yield of MABC-IL

NRCGCS-587, along with 12 elite cultivars, was tested for yield and related traits. The analysis of variance revealed significant differences among the genotypes and genotype × environment interaction for pod yield. In 2014 post-rainy season, pod yield of NRCGCS-587 was 1464 kg/ha that was significantly higher than the check cultivars Abhaya, CO-6, GG-20, ICGS-1043, JL-24, TMV-2 and VRI-6; on par with K-6, TAG-24 and GJG-31; and lower than TG-37A (Table 1). During 2015 rainy season, pod yield of NRCGCS-587 (1714 kg/ha) was significantly higher than check cultivars except for TG-37A and GG-20. The pooled pod yield of NRCGCS-587 (1589 kg/ha) was significantly higher than all the check cultivars except TG-37A. Shelling

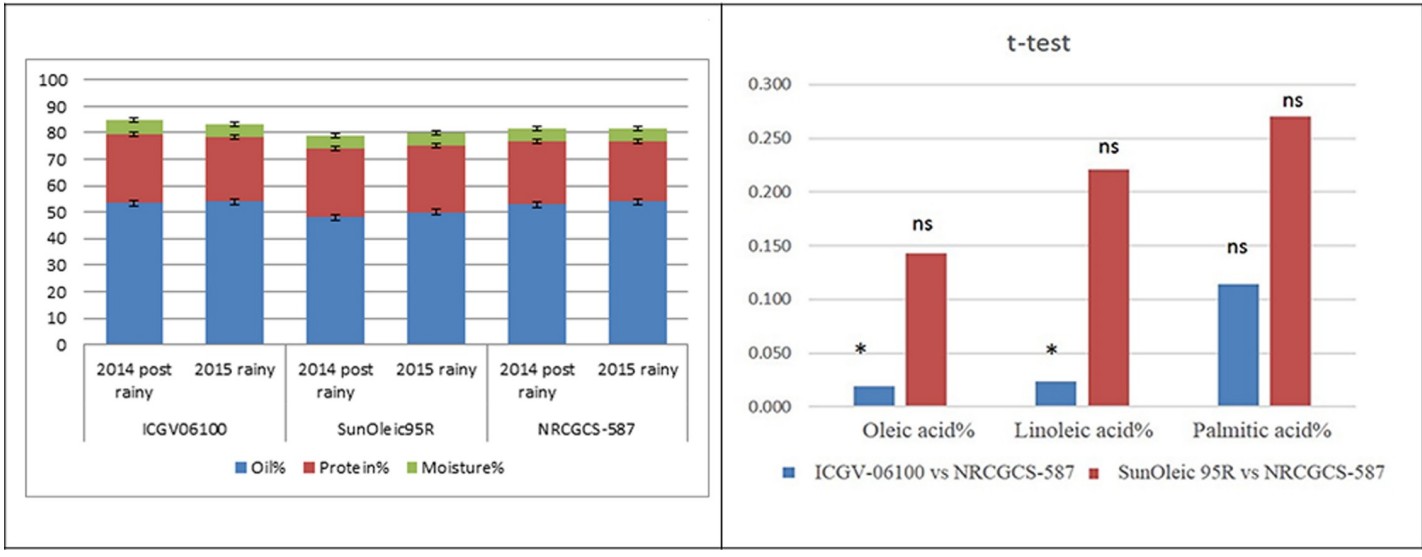

**Fig 4. Oleic acid, linoleic acid, and palmitic acid in NRCGCS-587 and parents grown in ICAR-DGR during 2014 post rainy and 2015 rainy; "*" indicates significance at 5%; "ns" indicates non-significant.**

percentage (73%) and hundred-kernel weight (50g) of NRCGCS-587 were higher with the check cultivars. Besides, NRCGCS-587 was tested at three different states over two seasons. AMMI analysis of variance (Table 2) revealed a significant interaction effect of genotype × location on pod yield followed by location and genotype, individually. Stability analysis in all the three locations by GGE biplot showed that pod yield of NRCGCS-587 was higher (Fig 6) with local check cultivars in Telangana (ICGS76) and Andhra Pradesh (TCGS-157) and superior to common check cultivar (GG-20).

## Oil content and fatty acid profile of MABC-IL in three different states

The pod samples of NRCGCS-587 were collected from three different states *viz.*, Andhra Pradesh, Telangana and Gujarat in 2016 post-rainy season and subjected to biochemical analysis

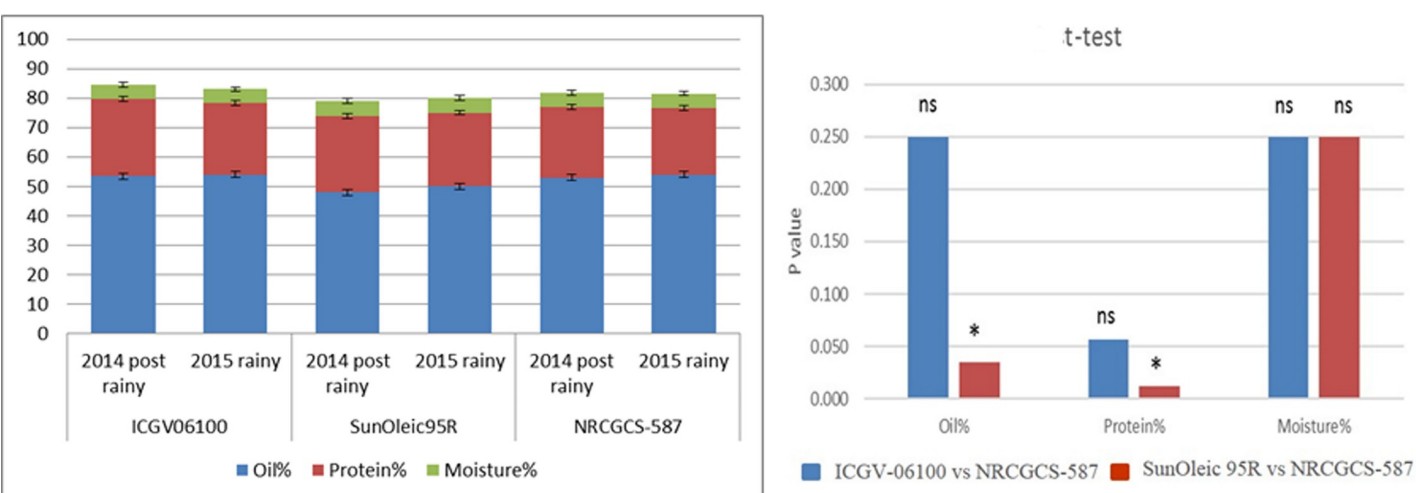

**Fig 5. Oil, protein, and moisture in NRCGCS-587 and parents grown in ICAR-DGR during 2014 post rainy and 2015 rainy seasons; "*" indicates significance at 5%; "ns" indicates non-significant.**

**Table 1. Yield and the related traits of NRCGCS-587 grown in ICAR-DGR, Gujarat, during 2014 post rainy and 2015 rainy season.**

| Genotypes | Pod Yield (kg/ha) | | | Shelling (%) | 100 kernel weight (g) |
|---|---|---|---|---|---|
| | **2015 rainy** | **2014 post rainy** | **Mean** | | |
| Abhaya | 1418.4 [c-d] | 1376.3 [b-d] | 1397.3 [b-d] | 72.1 [a-c] | 49.3 [b-d] |
| Co-6 | 1485.4 [c-d] | 1062.5 [d-e] | 1274.0 [b-e] | 70.7 [a-d] | 57.3 [a-b] |
| NRCGCS-587 | 1714.0 [b-c] | 1463.9 [b-c] | 1588.9 [b] | 72.1 [a-c] | 59.0 [a-b] |
| GG-20 | 1883.0 [a-b] | 967.9 e | 1425.4 [b-d] | 74 [a-b] | 65.7 [a] |
| GJG-31 | 1488.2 [c-d] | 1354.8 [b-e] | 1421.5 [b-d] | 66.4 [d] | 51.3 [b-d] |
| GPBD-4 | 1485.0 [c-d] | 1569.5 [b] | 1527.2 [b-c] | 74.1 [a-b] | 52.7 [b-d] |
| ICGS-1043 | 1450.8 [c-d] | 1178.5 [c-e] | 1314.7 [b-e] | 71.7 [a-c] | 54.3 [b-c] |
| JL-24 | 1385.6 [c-e] | 964.1 [e] | 1174.9 [c-e] | 69.7 [b-d] | 46.0 [c-e] |
| K-6 | 1336.5 [c-e] | 1499.4 [b-c] | 1417.9 [b-d] | 74.5 [a] | 53.3 [b-d] |
| TAG-24 | 1271.6 [d-e] | 1575.9 [b] | 1423.8 [b-d] | 70.9 [a-d] | 49.3 [b-d] |
| TG-37A | 2163.8 [a] | 2105.5 [a] | 2134.7 [a] | 70.4 [a-d] | 45.3 [c-e] |
| TMV-2 | 866.0 [f] | 1122.4 [c-e] | 994.2 [e] | 72.8 [a-c] | 44 [d-e] |
| VRI-6 | 1042.0 [e-f] | 1038.1 [d-e] | 1040.1 [d-e] | 68.7 [c-d] | 39.0 [e] |
| **CV%** | **15.08** | **14.97** | **14.28** | **13.90** | **18.61** |

Means followed by same letter are not significantly different (less than or equal) at P = 0.05.

(S2 Table). Oil content in NRCGCS-587 did not differ much across the states, i.e., 54.7%, 54.5%, and 55.1% in Telangana, Andhra Pradesh, and Gujarat, respectively. Oleic acid content was almost the same in the pods of the two states, *viz.*, Telangana (79.8%) and Andhra Pradesh (79.6%), while it was slightly higher in Gujarat (81.2%). Furthermore, linoleic acid (Telangana-3.0, Andhra Pradesh-3.5, and Gujarat-3.2) and palmitic acid contents (Telangana-6.5%, Andhra Pradesh-6.4%, and Gujarat-7.8%) across the locations were similar (Figs 7 and 8). Likewise, oleic to the linoleic ratio in NRCGCS-587 also remained almost the same.

## Passport data of NRCGCS-587 (MABC-IL) and recurrent parent

NRCGCS-587 is a Virginia bunch genotype characterized by decumbent-3 growth habit, alternate branching, green color, ovate leaf, and simple inflorescence. It takes about 23 days after

**Table 2. AMMI Analysis of variance for pod yield evaluated at the three locations.**

| | df | MSS | Pr (>F) | % Sum of Squares |
|---|---|---|---|---|
| Locations (L) | 5 | 4758501 | <0.001 | 36.8 |
| Rep (L) | 6 | 125004 | 0.22 | 1.2 |
| Genotype (G) | 9 | 968802 | <0.001 | 13.5 |
| G*L | 45 | 594607 | <0.001 | 41.3 |
| PC1 | 13 | 1174680 | 0 | 57.1 |
| PC2 | 11 | 539693.9 | 0 | 22.2 |
| PC3 | 9 | 349655.6 | <0.001 | 11.8 |
| PC4 | 7 | 241216.7 | 0.015 | 6.3 |
| PC5 | 5 | 142883.3 | 0.165 | 2.7 |
| Residuals | 54 | 87212 | | 7.3 |

PC1, PC2 . . .PC5 indicates principal components 1, 2. . ..5 (denotes variation accounted by each components); df–Degrees of freedom; MSS- Mean sum of squares. P- value at 5%.

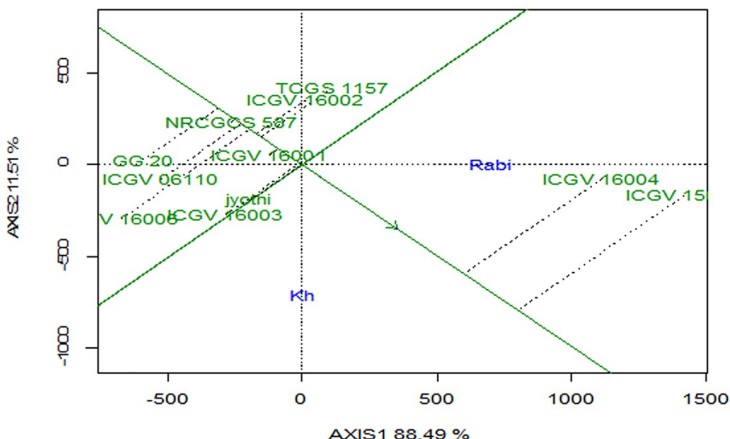

Andhra Pradesh (Tirupati)

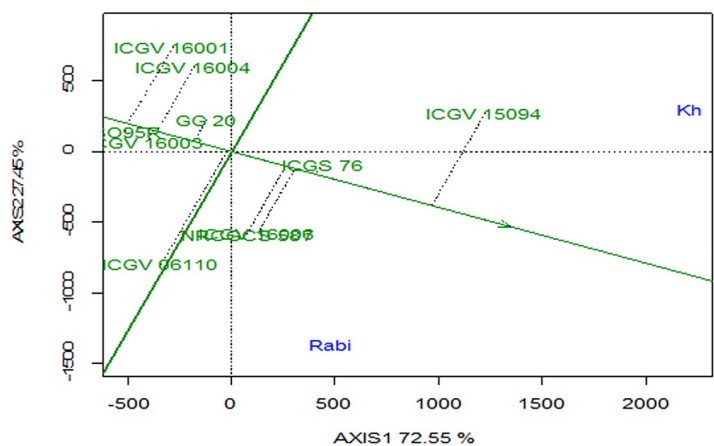

Telangana (ICRISAT)

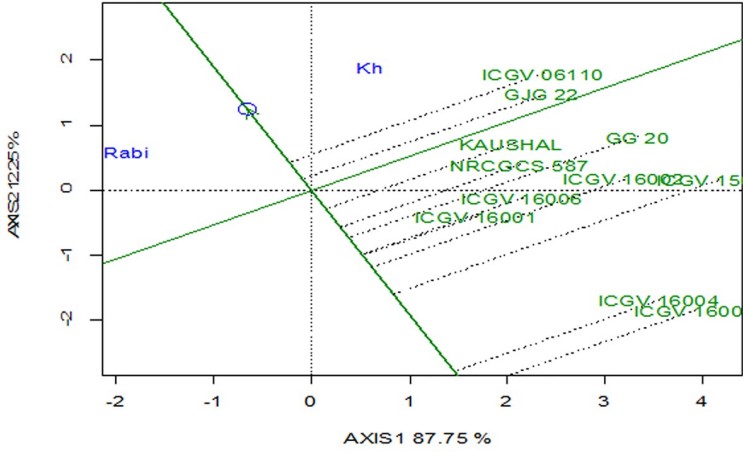

Junagadh

**Fig 6. Average environment coordination (AEC) views of the GGE-biplot based on environment-focused scaling peanut genotypes evaluated for pod yield in Andhra Pradesh, Telangana, and Gujarat, India.**

germination for 50% flowering and 115 days for maturity. Average plant height, leaf length and leaf width are 42.6 cm, 40.1 mm, and 13.2 mm, respectively. It produces an average of five primary branches per plant and 2–3 flowers per inflorescence. Pods are mostly two seeded and the average length and width of pods are 26.0 mm and 12.4 mm, respectively. The mean length and width of kernels are 13.8 mm and 6.8 mm, respectively and it is rose in color (Fig 9). It yields 108.0 g of pods per square meter with 20% harvest index, 70% shelling-out-turn, ~55% oil content,~80% oleic acid, and ~4% linoleic acid content (S3 Table). Most importantly, NRCGCS-587 has also shown resistance to rust and late leaf spot, i.e., 1 and 3 disease severity scores, respectively in 1–9 modified scale (data not shown).

## Seed and seedling traits

Average seed germination of 93.3% was found in normal oleic peanut, while it was 81.7% in HO peanut. A significant difference in germination percentage was recorded between normal and HO peanut (Table 3). There were no significant differences between normal and HO peanut for vigor index, fresh and dry plant weight, shoot and root length, fresh shoot and root weight, dry shoot and root weight, shoot length/root length, fresh shoot weight/fresh root weight, dry shot weight/dry root weight, and plant fresh weight/plant dry weight. However, the genotypic difference was observed within the normal and HO peanut groups. In both, the

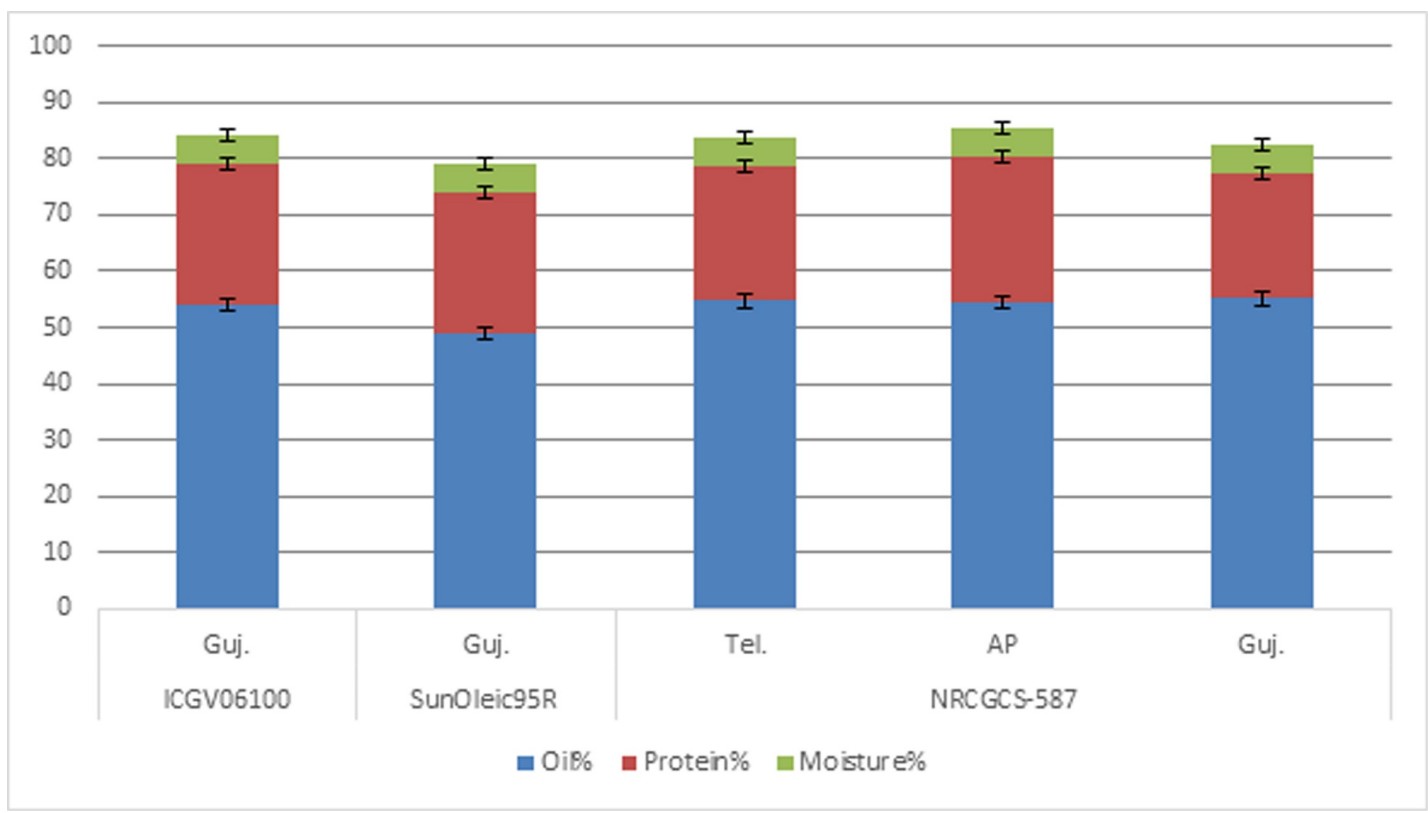

**Fig 7. Oil, protein, and moisture in NRCGCS-587 and parents grown in Andhra Pradesh, Telangana, and Gujarat, India during 2016 rainy season.**

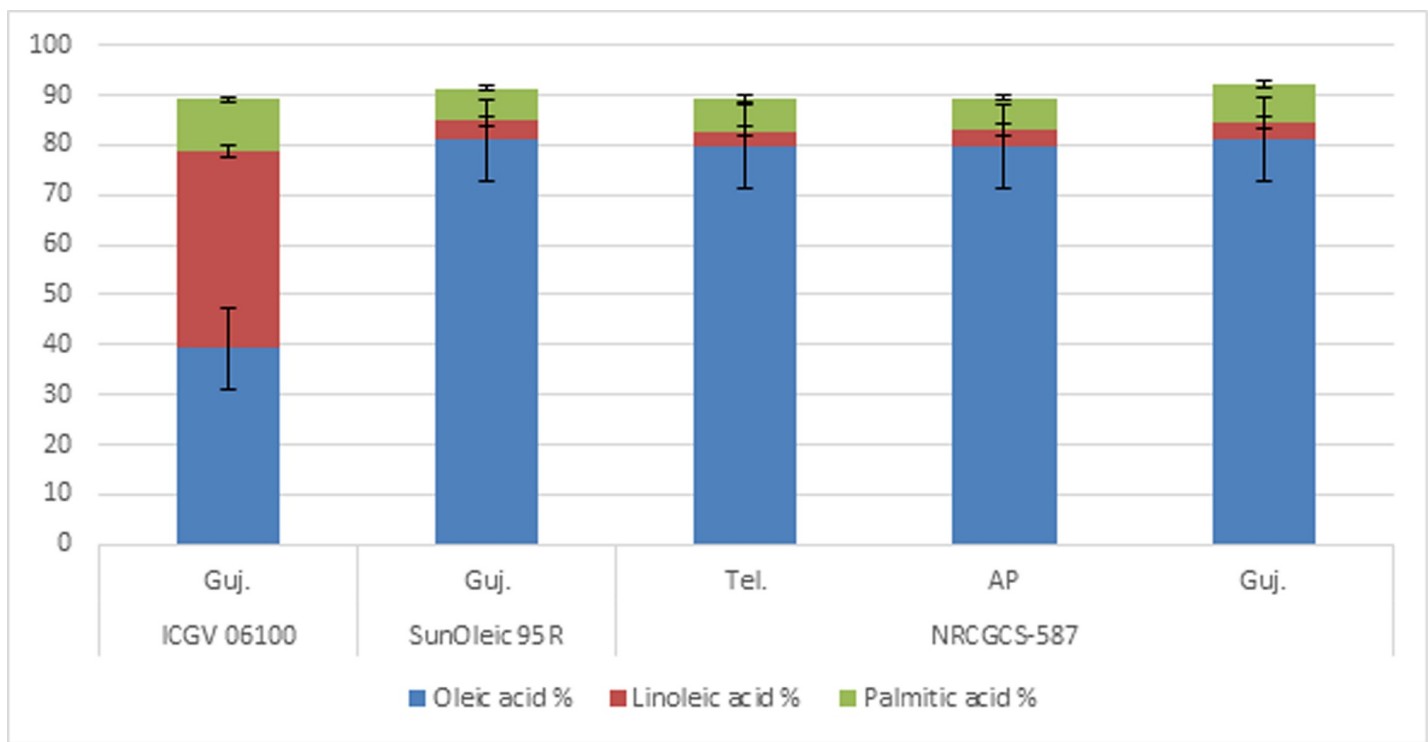

**Fig 8. Oleic acid, linoleic acid, and palmitic acid in NRCGCS-587 and parents grown in Andhra Pradesh, Telangana, and Gujarat, India during 2016 rainy season.**

groups shoot length, fresh shoot biomass, and dry shoot biomass were higher than fresh root length, fresh root biomass, and dry root biomass.

## Discussion

Peanut with HO is preferred over normal peanut due to its extended shelf life and multiple health benefits. High oil and oleic acid content in the peanuts are necessary for producing superior quality of oil to meet the nutritional needs and for industrial purposes. Moreover, the high oil containing peanuts can be used to combat malnutrition due to its higher caloric value over normal peanut. [48]. Therefore, improvement of oleic acid content in peanut for higher oxidative stability and better dietary properties is one of the important breeding objectives worldwide. Availability of molecular markers linked to the *ahFAD2* gene has facilitated marker-assisted breeding for HO. MABC breeding further ensures the transfer of desirable gene together with maximum genome recovery of the recurrent parent [49, 50]. Previously, nematode resistance [51], rust resistance [52], and high oleic acid [22, 23] traits were transferred to elite peanut cultivars using MABC breeding. The use of CAPS and SNP markers has considerably reduced the time and volume of breeding material in different backcross generations [25]. In the first objective, a high oil content peanut genotype, ICGV06100, was targeted to improve oleic acid content using MABC breeding. The studies reported the development of a peanut genotype, NRCGCS-587, with high oil and HO content. The HO trait was introgressed from SunOleic95R into the genetic background of ICGV06100 through MABC approach and developed an improved version of ICGV06100 with 97% increase in oleic acid content over the recurrent parent.

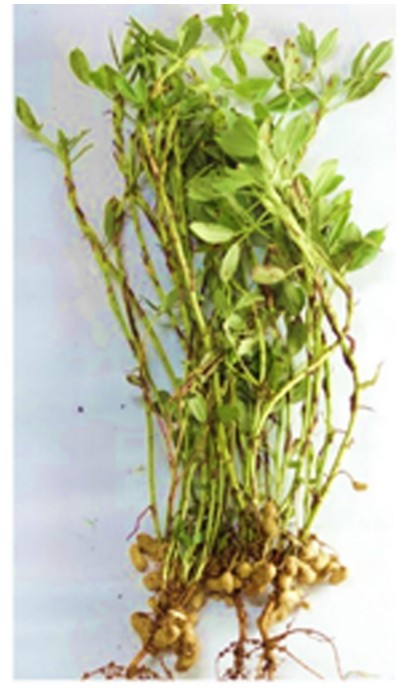

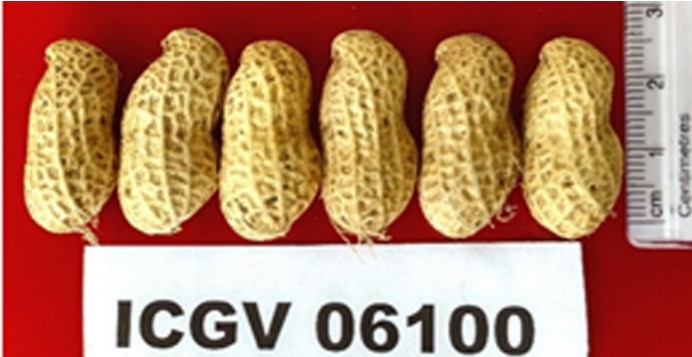

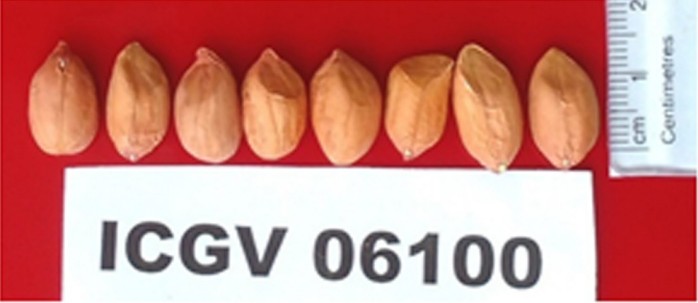

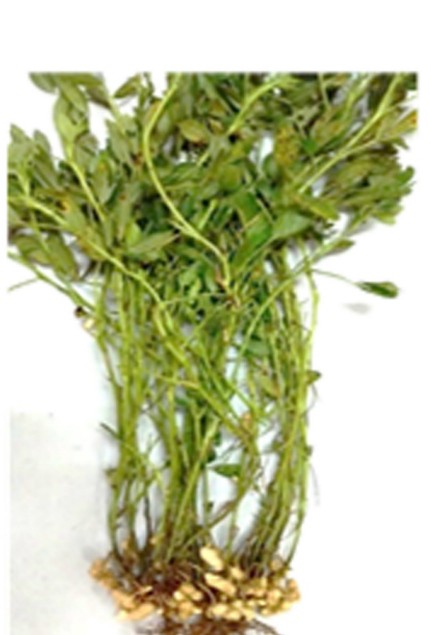

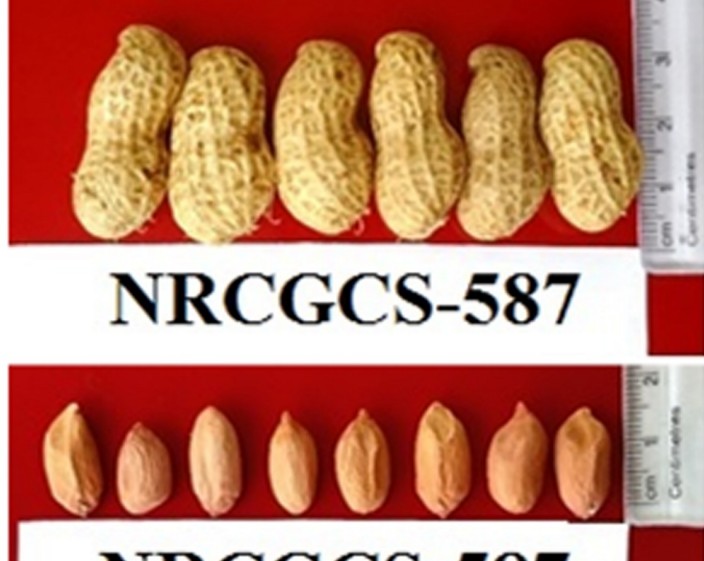

**Fig 9. Plant, pod, and kernels of ICGV06100 and NRCGCS-587.**

The increase in oleic acid content in NRCGCS-587 led to a reduction in linoleic acid. There was a 90% and 24% reduction in linoleic acid and palmitic acid, respectively, in NRCGCS-587 as compared to the recurrent parent. Moreover, linoleic acid content ranged from 3.0% to

Table 3. Details of seedling traits in normal oleic and high oleic peanut genotypes.

| Trait | Name of genotypes | Oil %* | Oleic acid %* | Germination % | Shoot Length (SL) | Root Length (RL) | SL/RL | Fresh Shoot wt. (FSW) (g) | Fresh Root wt. (FRW) (g) | FSW/FRW | Dry Shoot wt. (DSW) (g) | Dry Root wt. (DRW) (g) | DSW/DRW | Plant Fresh wt. (PFW) (g) | Plant Dry wt. (PDW) (g) | PFW/PDW | Vigor index |
|---|---|---|---|---|---|---|---|---|---|---|---|---|---|---|---|---|---|
| High oleic (~80%) peanuts | NRCGCS-587 | 55 | 80 | 80.00 | 17.78 | 7.65 | 2.34 | 1.97 | 0.16 | 11.94 | 0.32 | 0.05 | 6.90 | 2.13 | 0.37 | 5.77 | 0.3 |
| | HOP-IL_MAS-191 | 53.2 | 79.8 | 73.33 | 21.24 | 10.89 | 1.93 | 2.22 | 0.13 | 17.32 | 0.24 | 0.02 | 14.60 | 2.35 | 0.26 | 8.99 | 0.19 |
| | HOP-IL_MAS-145 | 54.5 | 80.3 | 76.7 | 23.00 | 9.55 | 2.40 | 2.56 | 0.15 | 17.03 | 0.32 | 0.01 | 26.55 | 2.71 | 0.33 | 8.23 | 0.25 |
| | HOP-IL_MAS-130 | 54.7 | 80.5 | 96.7 | 17.17 | 5.80 | 3.01 | 1.35 | 0.06 | 21.26 | 0.12 | 0.02 | 6.77 | 1.41 | 0.14 | 10.00 | 0.14 |
| | Mean | | | 81.70 | 19.79 | 8.47 | 2.42 | 2.02 | 0.13 | 15.98 | 0.25 | 0.02 | 13.71 | 2.15 | 0.28 | 7.81 | 0.22 |
| Normal oleic (~50–55%) peanuts | GG-20 | 51 | 64 | 90 | 22.40 | 6.75 | 3.32 | 2.58 | 0.12 | 22.43 | 0.28 | 0.03 | 10.24 | 2.70 | 0.31 | 8.82 | 0.27 |
| | ICGV-06100 | 55 | 39 | 83.30 | 15.46 | 5.93 | 2.63 | 2.02 | 0.35 | 5.77 | 0.27 | 0.05 | 6.36 | 2.37 | 0.32 | 7.38 | 0.26 |
| | ICGV-05141 | 54.7 | 55 | 100.00 | 14.68 | 6.63 | 2.28 | 1.24 | 0.08 | 15.48 | 0.14 | 0.01 | 10.37 | 1.32 | 0.16 | 8.32 | 0.16 |
| | ICGV-06110 | 53 | 38.3 | 100.00 | 17.20 | 4.15 | 4.42 | 1.62 | 0.06 | 29.36 | 0.10 | 0.01 | 20.40 | 1.67 | 0.11 | 15.61 | 0.11 |
| | Mean | | | 93.30 | 17.44 | 5.87 | 3.16 | 1.86 | 0.15 | 12.42 | 0.20 | 0.02 | 11.84 | 2.01 | 0.22 | 9.03 | 0.20 |
| | CD@5% | | | 7.55 | 5.16 | 1.42 | 0.9 | 0.73 | 0.04 | 3.65 | 0.1 | 0.01 | 5.51 | 0.76 | 0.1 | 2.57 | 0.08 |
| | CV% | | | 4.93 | 15.84 | 11.34 | 18.8 | 21.4 | 14.55 | 11.6 | 24.31 | 22.14 | 24.75 | 20.88 | 23.24 | 15.9 | 22.04 |

At 5% level of significance

*Source: [25, 26, 41]

4.0% and palmitic acid ranged from 6.1% to 7.8% over different locations indicating their stable expression. The O/L ratio was increased to 27 in NRCGCS-587 from 1.2 in the recurrent parent. A similar trend of increase in oleic acid and O/L ratio, as well as a reduction in linoleic acid and palmitic acid, has already been reported [22, 23]. Commonly, an alteration in any of the metabolite biosynthesis also has a negative feedback effect on the production of other metabolites in a related pathway. Likewise, a significant reduction in palmitic acid level in NRCGCS-587 was recorded. Several previous studies have also reported a similar effect of *ahFAD2* alleles on palmitic acid content [14, 22, 23, 53].

Generally, variation in oil content and fatty acid composition was reported in different environments due to the quantitative nature of these traits that are controlled by complex pathways [25, 26, 54]. However, limited or no variation was observed in NRCGCS-587 regarding oil, oleic, linoleic, and palmitic acid contents over locations indicating the minimal environmental effect on oil and HO traits. It seems that only a few independent genes, with the major effect, control oil and oleic acid production in NRCGCS-587. The selection for improved fatty acid composition would not affect the oil content of seed since there was no significant correlation between percent oil and any of the fatty acids or related variables [55]. Although fatty acid composition showed variation with the growth habit and environment, the oil content remained constant [56, 57, 58]. As a result, NRCGCS-587 with stable oil content across locations would be a better choice for use as a parent in the future breeding program on enhancing oleic acid and oil content in peanut.

NRCGCS-587 had more than 90% background genome recovery as well as precise introgression of *ahFAD2* alleles. Moreover, identical passport data of NRCGCS-587 and ICGV06100 except oleic acid content corroborate maximum genome recovery from recurrent parent and precise introgression of *ahFAD2* alleles in NRCGCS-587. Thus, NRCGCS-587 is an improved version of ICGV06100 having ~80% oleic acid content. The combined approach of both genotypic and phenotypic selections was found appropriate and effective in selecting improved lines [23, 59]. High oleic acid content did not affect seedling traits except the rate of germination. Significant variation in the rate of germination between HO and normal oleic peanut groups might be due to the alteration in lipid composition of seeds leading to changed membrane function and permeability. The germination decreased as O/L and unsaturated/saturated ratios increased in peanut, especially at lower (16°C and 14°C) temperatures [35]. Jungman and Schubert [36] reported that HO lines had lower seed vigor than their paired lines with normal oleic content. In general, the processes of germination initiates at a temperature below 15°C in peanut. Lower germination rate observed in HO peanut in this research might be due to the change in fatty acid composition since the temperature was maintained constant at 32°C. In sorghum, the α-amylase activity of seeds and subsequent seed germination percentage were affected by long-chain fatty acid composition [60].

In *Pinus pinea*, an increase in caprylic or oleic acids retarded the seed germination. The inhibition was dependent on fatty acid concentration and chain-length [61]. Short-chain fatty acids could infiltrate membrane lipids and change the physical properties that lower the seed germination [62].

In conclusion, there was a narrow but significant difference in seedling establishment between HO and normal oleic peanut under optimum temperature. Poor seed germination rate in HO peanut than normal peanut could be a cause of concern if a significant difference is more and needs further investigation to overcome it. A perfectly stable genotype having constant yield across geographical locations is a key to a successful variety [63]. The higher pod yield in the post-rainy season than a rainy season in NRCGCS-587 indicated that it might be more remunerative under irrigation than rain-fed conditions. It yielded either significantly higher or on par with all check cultivars except TG-37A indicating the potential to excel the

local elite varieties from different peanut-growing states in India. Shelling percent and hundred-kernel weight were also on par with elite cultivars. Furthermore, NRCGCS-587 recorded maximum pod yield (2445 kg/ha) in Telangana and Andhra Pradesh that makes it suitable for these states. Stable pod yield, oil content, and HO content of NRCGCS-587 over the locations make it more rewarding for the peanut growing farmers. NRCGCS-587 is an improved version of ICGV06100 having genotypically 91% RPG and *ahFAD2* alleles, and phenotypically high oil and yield. Thus, improved nutritional qualities would fetch premium price to the farmers without compromising the yield and meet the demand of peanut oil for industrial purposes.

## Supporting information

**S1 Table. Details of markers used in AS-PCR and CAPS analysis, background selection and testing of recombination in MABC line.**
(XLSX)

**S2 Table. Fatty acid profile MABC line and its parents.**
(DOCX)

**S3 Table. Qualitative, quantitative and special features of ICGV 06100 and NRCGCS-587 as per peanut descriptor.**
(DOCX)

## Acknowledgments

Authors acknowledge the financial support received from Integrated Scheme of Oilseeds, Pulses, Oilpalm and Maize (ISOPOM), Ministry of Agriculture and Farmers Welfare, the Government of India.

## Author Contributions

**Conceptualization:** Sandip K. Bera, Pasupuleti Janila, Manish K. Pandey, T. Radhakrishnan, Rajeev K. Varshney.

**Data curation:** Sandip K. Bera, Jignesh H. Kamdar, Mital D. Jasani, Kirti Rani.

**Formal analysis:** Swati V. Kasundra, Sahil V. Patel, Ajay B. Chandrashekar.

**Funding acquisition:** Pasupuleti Janila, T. Radhakrishnan, Rajeev K. Varshney.

**Investigation:** Sandip K. Bera.

**Methodology:** Sandip K. Bera, Jignesh H. Kamdar, Manish K. Pandey.

**Project administration:** Sandip K. Bera, Pasupuleti Janila, T. Radhakrishnan, Rajeev K. Varshney.

**Resources:** Rajeev K. Varshney.

**Software:** Jignesh H. Kamdar, Ajay B. Chandrashekar.

**Supervision:** Sandip K. Bera, A. K. Maurya, P. Dash, Rajeev K. Varshney.

**Validation:** Sandip K. Bera, Jignesh H. Kamdar, Sahil V. Patel, Mital D. Jasani, Kirti Rani.

**Visualization:** Sandip K. Bera.

**Writing – original draft:** Sandip K. Bera.

**Writing – review & editing:** Sandip K. Bera, Jignesh H. Kamdar, Ajay B. Chandrashekar, Kirti Rani, N. Manivannan, Pasupuleti Janila, Manish K. Pandey, R. P. Vasanthi, K. L. Dobariya, T. Radhakrishnan, Rajeev K. Varshney.

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
