## [Decision Letter · Decision Letter 0]

27 Aug 2019

PONE-D-19-15336

Steady expression of high oleic acid in peanut bred by marker-assisted backcrossing
for fatty acid desaturase mutant alleles and its effect on seed germination along
with other seedling traits

PLOS ONE

Dear Dr. Bera,

Thank you for submitting your manuscript to PLOS ONE. After careful consideration, we
feel that it has merit but does not fully meet PLOS ONE’s publication criteria as it
currently stands. Therefore, we invite you to submit a revised version of the
manuscript that addresses the points raised during the review process.

ACADEMIC EDITOR: **You will see that both the
reviewers made critical recommendations, but an important consideration is
whether the study describes a technically sound research and made significant
advances in the relevant area. However, as reviewers pointed out, I think that
this manuscript needs additional works. Therefore, I encourage the authors to
perform the revision as per the reviewer’s suggestions and if these were
meticulously performed, then I am sure that the MS could be reconsidered on a
later date.**

We would appreciate receiving your revised manuscript by Oct 11 2019 11:59PM. When
you are ready to submit your revision, log on to https://www.editorialmanager.com/pone/ and select the 'Submissions
Needing Revision' folder to locate your manuscript file.

If you would like to make changes to your financial disclosure, please include your
updated statement in your cover letter.

To enhance the reproducibility of your results, we recommend that if applicable you
deposit your laboratory protocols in protocols.io, where a protocol can be assigned
its own identifier (DOI) such that it can be cited independently in the future. For
instructions see: http://journals.plos.org/plosone/s/submission-guidelines#loc-laboratory-protocols

We look forward to receiving your revised manuscript.

Kind regards,

Manoj Prasad, PhD

Academic Editor

PLOS ONE

**Journal Requirements:**

**Comments to the Author**

1. Is the manuscript technically sound, and do the data support the conclusions?

Reviewer #1: Partly

Reviewer #2: Partly

2. Has the statistical analysis been performed appropriately and
rigorously? 

Reviewer #1: I Don't Know

Reviewer #2: No

3. Have the authors made all data underlying the findings in their
manuscript fully available?

Reviewer #1: Yes

Reviewer #2: No

4. Is the manuscript presented in an intelligible fashion and written
in standard English?

Reviewer #1: Yes

Reviewer #2: Yes

5. Review Comments to the Author

Reviewer #1: This manuscript was well written but it may not fit well to this
journal. I gave this recommendation for three clear reasons: 1. The authors tried to
describe a whole process for development of high oil and high oleate cultivars. The
experiment was not specifically designed for any genetic studies or chemical
analysis. 2. In the title, steady expression of high oleic acid and its effect on
seed germination along with other seedling traits did not match the manuscript
content. The evaluation of these traits was the selection outcome and did not
necessary related to high oleic acid. 3. Even before making the cross, authors did
not genotype parent ICGV06100 for FAD2 genotype. On Figure 2a and b, it was so
confused. The well images were so different, but they scored them into the same
genotype (for example, on Figure 2a well 6 and 7 to Aa; Figure 2b well 5, 6, and 7
images are the same but scored to different genotype Bb, BB, Bb). In addition,
Figure 1a, only 4 samples but the authors mentioned 1 to 5. Figure 1b image was not
clear. Furthermore, in the introduction (lines79-81), the authors mistook gadoleic
acid (C20:1) as saturated fatty acid.

Reviewer #2: Results:

Table 1 results are not included in the results section

Line 274 to 282 is this data represented in one of the figures or tables?

line 291 what is the p-values for Figure 3?

Line 294 p-value?, Line 296 25% protein contents, should state "with no significant
differences??"" and cite the p-value associated

Line 298 p-value

line 305 pod yield, p-value????

Line 311 shelling percentage in table 1 data, what are the stats and p-value??

Line 314 what are the p-values for the significant interaction effects??

Line 321 where is this data table???

Line 323 p-value is needed??

Line 328 p-value is needed to support statement

line 351 mention significant differences if any her in detail and in table 3, state
p-value

Line 361 to 362 please re-word for clarity is confusing as written

Table 1, table footnotes are needed, define importance of superscripts statistically,
what are the p-values

table 2 define in table footnote how yield mean sq was determined, describe in
footnotes the 3 locations, define PC1, etc, briefly state in footnotes stats
used

Table 3. table footnotes, table should in p-values, also, briefly state the methods
used, stats used, so that reader can more clearly understand the data

Figure 1 and Figure 2 are never discussed in the results section. also if these
figures are to be used they need to include figure legends with details of the
methods and quantity of the DNA starting materials for amplification

Figure 3, Figure 4, Figure 6. and Figure 7, figure legend is needed, p-values needed
with standard error bards, bar graphs using the current colors is difficult to read,
try black and white and different patterns, % of what?? % of total fatty acid,
x-axis values??

Figure 5???? not sure if this data adds to the strength of the manuscript??

Figure 8 same as previous comments of Figure 3, 4, also what are the harvest times
and from what three states, figure legend needed

Figure 9. this figure is never discussed in the manuscript. either include and
discuss in the results/discussion or remove. not sure if it adds to the strength of
the manuscript.

General

Line 71 and "the" rest

Line 73 and the "remaining on-third"

Line 74 low caloric (corrected spelling)

line 105, remove "in a while"

line 295 space after and

line 297 space after and

6. PLOS authors have the option to publish the peer review history of
their article (what does this mean?). If published, this will
include your full peer review and any attached files.

If you choose “no”, your identity will remain anonymous but your review may still be
made public.

**Do you want your identity to be public for this peer review?** For
information about this choice, including consent withdrawal, please see our
Privacy Policy.

Reviewer #1: No

Reviewer #2: No

---

## [Author Response · Author response to Decision Letter 0]

18 Sep 2019

Reviewer 1 

Comment 1: The authors tried to describe a whole process for development of high oil
and high oleate cultivars. The experiment was not specifically designed for any
genetic studies or chemical analysis

Reply 1: We accept the comment partially. 

Main aim of the experiment was to develop the introgression lines for high oleic
traits which we have been developed successfully.

Once we developed the introgression lines then only we aimed for genetics studies and
analysis of biochemical parameters and effect of high oleic acid on seed traits
which are in sequence one after another. Hope This properly justify our
experimentation. 

Comment 2: In the title, steady expression of high oleic acid and its effect on seed
germination along with other seedling traits did not match the manuscript content.
The evaluation of these traits was the selection outcome and did not necessary
related to high oleic acid.

Reply 2: We beg to differ from the comment of reviewer. The title of the manuscript
clearly describe three components of studies conducted by us.

Component 1: Steady expression of high oleic acid in peanut 

Component 2: (high oleic peanut) bred by marker-assisted backcrossing 

Component 3: (High oleic acid ) its effect on seed germination along with other
seedling traits

Hence, our manuscript exclusively describe the above three components and their
results. Hence, the title clearly match the manuscript.

Comment 3: 

Point-1: Even before making the cross, authors did not genotype parent ICGV 06100 for
FAD2 genotype. 

Point-2: On Figure 2a and b, it was so confused. The well images were so different,
but they scored them into the same genotype (for example, on Figure 2a well 6 and 7
to Aa; Figure 2b well 5, 6, and 7 images are the same but scored to different
genotype Bb, BB, Bb). In addition, Figure 1a, only 4 samples but the authors
mentioned 1 to 5. Figure 1b image was not clear.

Point 3: Furthermore, in the introduction (lines 79-81), the authors mistook gadoleic
acid (C20:1) as saturated fatty acid. 

Reply 3: We beg to differ from the comment of reviewer. 

Point-1: Genotyping of ICGV 06100 was very much included in the gel picture (Figure
2a and 2b). In the figure 2a and 2b lane marked as P1 depicts the recurrent parent-1
which is ICGV 06100. However, we regret to mention here that labelling in the figure
2a and 2b was not proper which we have corrected subsequently.

Point-2: We wholly accept the comments of reviewer for mismatching the labelling and
quality of Fig. 1a, 1b 2a and 2b.

Fig (1a, 1b, 2a and 2b) have been replaced by clear gel pictures with proper and
corrected labelling. Also scoring details of these gel pictures have been added in
Materials and methods (DNA extraction and marker genotyping subheading) to make the
scoring clear.

Point 3: Necessary correction has been made in the manuscript.

Reviewer 2 

Comments 1: Table 1 results are not included in the results section

Reply 1: We accept the comment of reviewer. We unknowingly made mistake to mention
table 1 in the manuscript which we have included under subheading “Recurrent parent
genome recovery and linkage drag” of result section.

Comment 2: Line 274 to 282 is this data represented in one of the figures or
tables?

Reply 2: We accept the comment of reviewer. We have made necessary correction and
mentioned “Table 1” in the “Recurrent parent genome recovery and linkage drag” of
result section 

Comment 3: Line 291 what is the p-values for Figure 3?

Reply 3: We accept the comment of reviewer. Necessary correction has been included as
P value at 5 % level of significance (added in footnote of Fig.3)

Comment 4: Line 294 p-value?, Line 296 25% protein contents, should state "with no
significant differences??"" and cite the p-value associated 

Reply 4: We accept the comment of reviewer. Necessary correction (one sentence of no
significant difference as depicted in figure 4 has been added in manuscript). P
value at 5 % level of significance (added in footnote of Fig.4) 

Comment 5: Line 298 p-value

Reply 5: We accept the comment of reviewer. p-value is at 5 % level of significance
(added in footnote in Fig.3)

Comment 6: line 305 pod yield, p-value????

Reply 6: We accept the comment of reviewer. p-value at 5 % level of significance
added in the table 1.

Comment 7: Line 311 shelling percentage in table 1 data, what are the stats and
p-value?? 

Reply 7: We accept the comment of reviewer. We have used Duncan’s Multiple Range Test
(DMRT) for test of significance at 5%.

Comment 8: Line 314 what are the p-values for the significant interaction effects?? 

Reply 8: We accept the comment of reviewer. Significance at 5% included in table
2.

Comment 9: Line 321 where is this data table??? Line 323 p-value is needed?? Line 328
p-value is needed to support statement.

Reply 9: We accept the comment of reviewer. Results described in the results section
is completely based on figure 6 and figure 7. We depicted the data in bar diagram
only not mentioned in a fresh table. The numerical values of oil content, Oleic,
Linoleic and palmitic acids between states were more or less similar. Hence neither
we compared it statistically nor added as a separate table. Hence, no p-value is
mentioned.

Comments 10: Line 351 mention significant differences if any her in detail and in
table 3, state p-value 

Reply 10: We accept the comment of reviewer. P-vale at 5% included in the Table
3.

Comment 11: Line 361 to 362 please re-word for clarity is confusing as written 

Reply 11: We accept the comment of reviewer. We have edited the sentence to make to
make meaning clear 

Comment 12: Table 1, table footnotes are needed, define importance of superscripts
statistically, what are the p-values 

Reply 12: We accept the comment of reviewer. We have made necessary correction in the
Table 1.

Comment 13: Table 2 define in table footnote how yield mean sq was determined,
describe in footnotes the 3 locations, define PC1, etc, briefly state in footnotes
stats used 

Reply 13: We accept the comment of reviewer. We have made necessary correction in the
Table 2. Name of three locations were mentioned in the write up, hence avoid in the
table to avoid clumsiness.

Comment 14: Table 3. table footnotes, table should in p-values, also, briefly state
the methods used, stats used, so that reader can more clearly understand the data 

Reply 14: We accept the comment of reviewer. Footnotes added in the table 3.
Methodology and stats used have been mentioned in the material methods section.

Comment 15: Figure 1 and Figure 2 are never discussed in the results section. also if
these figures are to be used they need to include figure legends with details of the
methods and quantity of the DNA starting materials for amplification 

Reply 15: We accept the comment of reviewer. Results of Figure 2 was already
discussed in the manuscript. However, results of Figure 1 was missing. We have added
the results of Figure 1(with legends) in Materials and methods (DNA extraction and
marker genotyping), methods of genotyping has also been added and references have
been added.

Comment 16: Figure 3, Figure 4, Figure 6. and Figure 7, figure legend is needed,
p-values needed with standard error bards, bar graphs using the current colors is
difficult to read, try black and white and different patterns, % of what?? % of
total fatty acid, x-axis values?? 

Reply 16: We accept the comment of reviewer. We have modified all the graphs
accordingly. X- axis is self-explanatory (indicates % of total fat)

Comment 17: Figure 5???? not sure if this data adds to the strength of the
manuscript??

Reply 17: We accept the comment of reviewer. Yes, Fig. 5 is needed as it indicates
stability and high yield of NRCGCS-ILs over locations by GGE biplot analysis and to
identify such genotype was ours second objective.

Comment 18: Figure 8 same as previous comments of Figure 3, 4, also what are the
harvest times and from what three states, figure legend needed

Reply 18: We accept the comment of reviewer. Figure 8 is essential, as it shows the
passport data of new identified line NRCGCS-587 (COMPARED WITH RECURRENT PARENT)

One sentence on harvest time i.e., Harvest on maturity of crop (110-115 days after
sowing) has been added in Yield evaluation section of MABC-ILs in results
section.

Comment 19: Figure 9. this figure is never discussed in the manuscript. either
include and discuss in the results/discussion or remove. not sure if it adds to the
strength of the manuscript.

Reply 19: We accept the comment of reviewer. Figure 9 has now been added in materials
and methods part (seed and seedling traits), figure is a part of explanation of
methodology followed to grow the seedlings to screen for important morphological
traits to characterize the genotype.

Comment 20: General comments

Line 71 and "the" rest

Line 73 and the "remaining on-third"

Line 74 low caloric (corrected spelling)

line 105, remove "in a while"

line 295 space after and

line 297 space after and 

Reply 20: We accept the comments of reviewer. We have incorporated all the
corrections highlighted by the reviewer.

to Reviewers.docx
---

## [Decision Letter · Decision Letter 1]

5 Nov 2019

PONE-D-19-15336R1

Steady expression of high oleic acid in peanut bred by marker-assisted backcrossing
for fatty acid desaturase mutant alleles and its effect on seed germination along
with other seedling traits

PLOS ONE

Dear Dr. Bera,

Thank you for submitting your manuscript to PLOS ONE. After careful consideration, we
feel that it has merit but does not fully meet PLOS ONE’s publication criteria as it
currently stands. Therefore, we invite you to submit a revised version of the
manuscript that addresses the points raised during the review process.

**Thank you for submitting your revised manuscript. You will see that one of the
reviewers make critical recommendations and it appears that additional work is
needed as indicated by the reviewers. If these were meticulously performed, then
I am sure that the MS could be reconsidered on a later date.**

We would appreciate receiving your revised manuscript by Dec 20 2019 11:59PM. When
you are ready to submit your revision, log on to https://www.editorialmanager.com/pone/ and select the 'Submissions
Needing Revision' folder to locate your manuscript file.

If you would like to make changes to your financial disclosure, please include your
updated statement in your cover letter.

To enhance the reproducibility of your results, we recommend that if applicable you
deposit your laboratory protocols in protocols.io, where a protocol can be assigned
its own identifier (DOI) such that it can be cited independently in the future. For
instructions see: http://journals.plos.org/plosone/s/submission-guidelines#loc-laboratory-protocols

We look forward to receiving your revised manuscript.

Kind regards,

Manoj Prasad, PhD

Academic Editor

PLOS ONE

Reviewers' comments:

Reviewer's Responses to Questions

**Comments to the Author**

1. If the authors have adequately addressed your comments raised in a previous round
of review and you feel that this manuscript is now acceptable for publication, you
may indicate that here to bypass the “Comments to the Author” section, enter your
conflict of interest statement in the “Confidential to Editor” section, and submit
your "Accept" recommendation.

Reviewer #2: All comments have been addressed

Reviewer #3: All comments have been addressed

Reviewer #4: (No Response)

2. Is the manuscript technically sound, and do the data
support the conclusions?

Reviewer #2: Yes

Reviewer #3: Yes

Reviewer #4: Partly

3. Has the statistical analysis been performed
appropriately and rigorously? 

Reviewer #2: Yes

Reviewer #3: Yes

Reviewer #4: Yes

4. Have the authors made all data underlying the
findings in their manuscript fully available?

Reviewer #2: Yes

Reviewer #3: Yes

Reviewer #4: Yes

5. Is the manuscript presented in an intelligible
fashion and written in standard English?

Reviewer #2: Yes

Reviewer #3: Yes

Reviewer #4: Yes

6. Review Comments to the Author

Reviewer #2: (No Response)

Reviewer #3: The authors have responded to all questions raised by the previous
reviewer except the question about the title. They need to change the title because
it does not reflects the contents of manuscript.The title should mention the main
achievement which is the development of high oleic content line. The title might
read something like this "Development of stable lines with high oleic content
through marker assisted introgression of fatty acid desaturate mutant alleles and
its effect on seed germination". All other questions raised by previous reviewer has
been answered and have improved the quality of manuscript significantly.

Reviewer #4: I feel that the title may be modified. Currently, the title seems to
indicate that variations in germination frequencies and other seedling traits are
primarily due to high oleic acid, which may not be the case due to various reasons
including genomic reconstitution as elaborated below. It would be advisable to list
these as separate features/studies of the high oleic lines.

In the abstract, the authors talk about resistance to biotic stresses but there is no
mention of this data in the manuscript. This portion towards the end should be
deleted.

Line 66: "in around"

I have the following four important queries which the authors need to clarify:

1. Lines 191-195: Aren't four SSRs per linkage group too less for estimating
recurrent genome reconstitution?

2. Why was backcrossing done only till BC3? Would it be sufficient to reconstitute a
significant portion of the recurrent parent genome in the absence of background
selection.

Since background selection was not done, in conjunction to point 1 above, there could
be sufficient portions of the non-recurrent parent genome remaining in the resulting
selected line which could be responsible for other variations observed in seedling
traits. Also, this would raise questions about the stability of the generated
line.

3. Was stable inheritance of the high oleic phenotype tested using successive
generations of harvested seed? This is not clearly given in the text. How was it
done? Which generations of selfed progeny were used?

4. How is passport data a good approach to test for genomic reconstitution? I am not
in agreement with this approach.

Line 226: Shouldn't it be "111-115"?

Line 260: Any references for this formula?

Line 284: Should be "codified" or "named" or "termed" instead of "decoded"

Line 290: (RPG) in brackets

Line 292: What does "nine were amplified" imply? Did they harbor the SunOleic95R
genome segments? Is there any information available on the genetic/physical distance
of these 10 SSRs?

Line 368-369: "However, the..........the groups" is not clear. Rephrase.

Figure 1 legend: Please correct for language

7. PLOS authors have the option to publish the peer
review history of their article (what does this mean?). If published, this will
include your full peer review and any attached files.

If you choose “no”, your identity will remain anonymous but your review may still be
made public.

**Do you want your identity to be public for this peer review?** For
information about this choice, including consent withdrawal, please see our
Privacy Policy.

Reviewer #2: No

Reviewer #3: No

Reviewer #4: No

---

## [Author Response · Author response to Decision Letter 1]

12 Nov 2019

Reviewer #3: 

Comments: 

The authors have responded to all questions raised by the previous reviewer except
the question about the title. They need to change the title because it does not
reflects the contents of manuscript. The title should mention the main achievement
which is the development of high oleic content line. The title might read something
like this "Development of stable lines with high oleic content through marker
assisted introgression of fatty acid desaturate mutant alleles and its effect on
seed germination". All other questions raised by previous reviewer has been answered
and have improved the quality of manuscript significantly. 

Reply:

We beg to differ from the comment made by the reviewer #3. The reviewer #3 mentioned
that the authors had not responded to the question about the change of title which
is absolutely incorrect. During the first review reviewer #1 offered the same
comment in his comment no.-2. In reply we submitted justification for not making any
changes in the title of our manuscript (Can be referred earlier review). We still
stand with our justification this time also and do not see any point for making
changes in the current title of our manuscript. However, final decision lies with
the editor.

Reviewer #4: 

Comment 1: 

I feel that the title may be modified. Currently, the title seems to indicate that
variations in germination frequencies and other seedling traits are primarily due to
high oleic acid, which may not be the case due to various reasons including genomic
reconstitution as elaborated below. It would be advisable to list these as separate
features/studies of the high oleic lines.

Reply:

We beg to differ from the comment of the reviewer. The title of the manuscript
clearly describe three components of the studies conducted by us.Component 1: Steady
expression of high oleic acid in peanut; Component 2: Breeding of high oleic peanut
by marker-assisted backcrossing; 

Component 3: Effect of high oleic acid on seed germination along with other seedling
traits. Hence, our manuscript clearly describe the above three components and their
results. Hence, the title clearly match the manuscript.

Comment 2: 

In the abstract, the authors talk about resistance to biotic stresses but there is no
mention of this data in the manuscript. This portion towards the end should be
deleted

Reply:

The data on resistance to biotic stresses (Rust and Late Leaf Spot diseases) of
NRCGCS-587 and ICGV 06100 (recurrent parent) have been mentioned in supplementary
Table–2 (S2). It may be treated as additional information. Hence, deleting the
statement on resistance to biotic stresses from the text may not be required.

Comment 3:

Line 66: "in around"

Reply:

Accepted, Necessary correction has been made in the manuscript.

Comment 4:

Lines 191-195: Aren't four SSRs per linkage group too less for estimating recurrent
genome reconstitution?

Reply:

It is always better to use SSRs as many as possible for estimating recurrent genome
reconstitution. However, we selected four anchored SSRs from each linkage group
(altogether 80 SSRs) which are genetically mapped in the peanut consensus map. Out
of which 79 SSRs gave similar expression both in recurrent parent and introgression
line. It gives a fair indication about the reconstitution of the recurrent parent
genome in introgression line. Furthermore, we used three cycles of backcrossing
which essentially (mathematically) recovers 93.75% of the recurrent parent genome.
Both these approaches are supplementary to each of above two approaches and confirm
recurrent parent genome reconstitution. 

Comment 5:

Why was backcrossing done only till BC3? Would it be sufficient to reconstitute a
significant portion of the recurrent parent genome in the absence of background
selection. Since background selection was not done, in conjunction to point 1 above,
there could be sufficient portions of the non-recurrent parent genome remaining in
the resulting selected line which could be responsible for other variations observed
in seedling traits. Also, this would raise questions about the stability of the
generated line.

Reply:

Theoretically, three cycle of backcrossing supposed to recover 93.75% recurrent
parent genome. We attempted three cycles of backcrossing which helped us to recover
93.75% recurrent parent genome. In such case taking background selection
additionally would not yield any more information. However, one can do any number of
additional work to strengthen the result. Yes, there is a chance of having 6.25% of
non-recurrent parent genome in the introgression line and literally, this 6.25% of
non-recurrent parent genome may throw little or no variations among the
introgression lines. 

Comment 6:

3. Was stable inheritance of the high oleic phenotype tested using successive
generations of harvested seed? This is not clearly given in the text. How was it
done? Which generations of selfed progeny were used?

Reply:

Yes, we have tested the high oleic phenotype in F3, F4 and F5 generations and
confirmed. 

It is clearly mentioned in the text line no. 300 (for the year 2014), 308 (for the
year 2015) and 337 (for the year 2016 in multiplication).

High oleic trait was phenotyped through fatty acid analysis of kernels in Gas
chromatography (Details methodology have been described in Materials and Methods
section).

Selfed progenies of F3, F4 and F5 generations.

comment 7: 

How is passport data a good approach to test for genomic reconstitution? I am not in
agreement with this approach.

Reply:

Passport data of a genotype/verities of a particular crop is an important information
which is used in DUS (Distinctness, Uniformity and Stability) testing. It helps to
distinguish/identify one genotype from others. If two genotypes are similar in
majority of the passport traits except one or two major trait(s), the genotypes can
also be refereed as near isogenic line (NIL). Here, recurrent parent and
introgression line are similar in majority of their passport traits except the high
oleic content which we have introgressed into the introgression line. Theoretically,
3 cycle of backcrossing supposed to recover 93.75% genome of recurrent parent. Thus
introgression line, NRCGCS-587 has got more than 90% genome constitution of
recurrent parent (ICGV 06100), which has been reflected in the passport data of
introgression line and recurrent parent.

Comment 8:

Line 226: Shouldn't it be "111-115"?

Reply:

Accepted, Necessary correction has been made in the manuscript.

Comment 9:

Line 260: Any references for this formula?

Reply:

Accepted, Necessary correction has been made in the manuscript.

Comment 10:

Line 284: Should be "codified" or "named" or "termed" instead of "decoded"

Reply:

Accepted, Necessary correction has been made in the manuscript.

Comment 11:

Line 290: (RPG) in brackets

Reply:Accepted, Necessary correction has been made in the manuscript.

comment 12:

Line 292: What does "nine were amplified" imply? Did they harbor the SunOleic95R
genome segments? Is there any information available on the genetic/physical distance
of these 10 SSRs?

Reply:

Accepted, Necessary correction has been made in the manuscript.

Yes, there is always possibilities, possible explanation is mentioned in the text. 

Yes, Necessary information is already available in the supplementary table 1
(S1).

Comment 13:

Line 368-369: "However, the..........the groups" is not clear. Rephrase.

Reply:

Accepted, Necessary correction has been made in the manuscript.

Comment 14:

Figure 1 legend: Please correct for language

Reply:

Accepted, Necessary correction has been made in the manuscript.

to Reviewers.docx
---

## [Decision Letter · Decision Letter 2]

25 Nov 2019

Steady expression of high oleic acid in peanut bred by marker-assisted backcrossing
for fatty acid desaturase mutant alleles and its effect on seed germination along
with other seedling traits

PONE-D-19-15336R2

Dear Dr. Bera,

We are pleased to inform you that your manuscript has been judged scientifically
suitable for publication and will be formally accepted for publication once it
complies with all outstanding technical requirements.

With kind regards,

Manoj Prasad, PhD

Academic Editor

PLOS ONE

Additional Editor Comments (optional):

Reviewers' comments:

Reviewer's Responses to Questions

**Comments to the Author**

1. If the authors have adequately addressed your comments raised in a previous round
of review and you feel that this manuscript is now acceptable for publication, you
may indicate that here to bypass the “Comments to the Author” section, enter your
conflict of interest statement in the “Confidential to Editor” section, and submit
your "Accept" recommendation.

Reviewer #3: All comments have been addressed

Reviewer #4: All comments have been addressed

2. Is the manuscript technically sound, and do the data
support the conclusions?

Reviewer #3: Yes

Reviewer #4: (No Response)

3. Has the statistical analysis been performed
appropriately and rigorously? 

Reviewer #3: Yes

Reviewer #4: (No Response)

4. Have the authors made all data underlying the
findings in their manuscript fully available?

Reviewer #3: Yes

Reviewer #4: (No Response)

5. Is the manuscript presented in an intelligible
fashion and written in standard English?

Reviewer #3: Yes

Reviewer #4: (No Response)

6. Review Comments to the Author

Reviewer #3: I am surprised at stubborn attitude of authors. Three different
reviewers have pointed out discrepancy in manuscript content and the title. The
reasons have been cited and alternate titles have been suggested. Reviewers have
been constructive in reviewing the manuscript but authors are reiterating the same
comment again and again. They have not even considered what reviewers have been
pointing out. Under these circumstances, I have no other comments to make and leave
it to wisdom of editor to decide what will be right. I will not like to take up this
manuscript for reviewing anymore.

Reviewer #4: (No Response)

7. PLOS authors have the option to publish the peer
review history of their article (what does this mean?). If published, this will
include your full peer review and any attached files.

If you choose “no”, your identity will remain anonymous but your review may still be
made public.

**Do you want your identity to be public for this peer review?** For
information about this choice, including consent withdrawal, please see our
Privacy Policy.

Reviewer #3: No

Reviewer #4: No

---

## [Editor Report · Acceptance letter]

5 Dec 2019

PONE-D-19-15336R2 

Steady expression of high oleic acid in peanut bred by marker-assisted backcrossing
for fatty acid desaturase mutant alleles and its effect on seed germination along
with other seedling traits 

Dear Dr. Bera:

I am pleased to inform you that your manuscript has been deemed suitable for
publication in PLOS ONE. Congratulations! Your manuscript is now with our production
department. 

With kind regards,

on behalf of

Dr. Manoj Prasad 

Academic Editor

PLOS ONE